# The relationship between screen time and gross motor movement: A cross-sectional study of pre-school aged left-behind children in China

Rui Yuan[1], Jia Zhang[2]*, Pengwei Song[3], Long Qin[3]

1 Department of Physical Education, University of Shanghai for Science and Technology, Shanghai, China, 2 School of Physical Education, Chongqing University, Chongqing, China, 3 School of Physical Education, Guangxi Science and Technology Normal University, Laibin, Guangxi, China

* zhangjiaaa@cqu.edu.cn

## Abstract

### Objective

To investigate the level of screen time and gross motor movement level and the correlation between them in left-behind children aged 3 to 6 years old in China.

### Methods

A randomized whole-group sampling method was used to study 817 left-behind children aged 3–6 years in 15 kindergartens in Xiangcheng city, Henan province. The third version of the Test of Gross Motor Development (TGMD-3) was used to test the children's gross motor movement level, and the screen time questionnaire was used to test the children's screen time level. The relationship between the two and the indicators was explored using Pearson's two-sided correlation and multilevel regression.

### Results

The average daily screen time of left-behind children aged 3–6 years old increased with age, and the reporting rate of >2 h/d ranged from 22.43% to 33.73%; gross motor movement of left-behind children aged 3–6 years old increased with age, with significant differences between age ($p<0.05$). There was a low to moderate negative correlation ($r = -0.133$ to $-0.354$, $p<0.05$) between screen time and gross motor movement in children aged 3–6 years, and multiple regression analysis showed that screen time was predictive of gross motor movement in children ($p<0.05$), with an explanation rate of 21.4%.

### Conclusion

There is a correlation between screen time and gross motor movement development in children aged 3–6 years old left behind, and the gross motor movement ability of children aged 3–6 years old can be developed by reducing screen time and increasing physical activity.

**Data Availability Statement:** All relevant data are within the paper.

**Funding:** This work was supported by Fundamental Research Funds for the Central Universities (2023CDSKXYTY003). The funders had no role in study design, data collection and analysis, decision to publish, or preparation of the manuscript.

**Competing interests:** The authors have declared that no competing interests exist.

## Introduction

Left-behind children are children under 16 years old who have been left behind in their rural hometowns when their parents migrate elsewhere to work [1]. Parental labor migration is common in some developing countries (China, South and South East Asia, and Africa), causing many children to be left behind [2, 3]. Henan is a large province of left-behind children; as of 2021, 1,595,600 rural left-behind children in compulsory education are in school, accounting for 10.70% of the total number of students in compulsory education, excluding preschool left-behind children (More than 120,000) [4]. In 2016, the government promulgated the "Opinions on Strengthening the Care and Protection of Children Left Behind in Rural Areas," The "Opinions" require a sound mechanism and strengthened supervision to solve the problem of children left behind by 2022. Building a management system and education for left-behind children is an important task.

Preschool left-behind children mean left-behind children in preschool education from 3 to 6 years old. This period is the essential stage of children's motor skills acquisition and the best time to become proficient in stability, mobility, and movement skills [5]. At this time, if the foundation of motor skills is laid, developing body movement habits can bring lifelong benefits [6, 7]. During the 3–6 years of age, children's gross motor movement reaches a primary stage where children's coordination and level of motor control are improved to some extent [8, 9]. If children are given enough motor skill practice and instruction at this time, they can develop early motor experiences and promote the development of gross motor movement. The 3–6 year old stage is critical for essential motor development [10]. Gross motor movements are the basis for later high motor and special motor skills. They are an essential way to explore the environment later and learn about the world around them. If children do not acquire the correct basic movement patterns early in life, their ability to acquire different combinations of gross motor movement will be significantly reduced [11, 12]. Therefore, optimal development of these skills is essential for the developing child.

The development of gross motor movement in early childhood mainly includes the separate locomotor and the action behavior of manipulating objects, which is an integral part of fundamental motor skills [13]. Separate locomotor mainly refers to the movement of the body in space, including running, jumping, crossing, sliding, fast and sharp stopping, etc. 3–6 years old is the peak and sensitive period of movement development, and many gross motor movement development changes a lot before the age of 8 [14]. Therefore, focusing on the development of gross motor movements such as walking, running, jumping, and throwing from the early childhood stage is very important for the mastery of advanced motor skills in the future. Furthermore, research has shown that gross motor movement enables children to achieve their goals of independent activity and exploration, which benefits their physical and mental health and social adaptability. Studies have confirmed that gross movement is closely related to physical health and cognitive development [15–17]. Motor development is essential for monitoring, evaluating, and diagnosing individuals' physical and mental development [18]. Gross motor movements can predict physical activity participation, physical fitness levels, and even cognitive performance in adulthood [19].

Motor development theory suggests that gross motor movement is stage-specific and programmatic as one of the critical components of movement development [20]. From infancy to adolescence, gross motor development increases with age [21]. However, it has been shown that many children need to obtain mastery levels of competency in many gross motor skill test items, especially object control and ball handling skills such as throwing, catching, and striking [22]. This affects their performance in athletic endeavors, but more importantly, it may affect their affinity for moderate-to-vigorous physical activity [23]. Therefore, intervention and

training of fundamental movement skills at 3 to 6 years old can facilitate children to reach a higher level of movement in their future daily lives. Therefore, it is vital to grasp the golden period of children's learning to do systematic and efficient movement, play, and learning.

The Internet plays an increasingly important role in the development of children and adolescents [24]. While optimizing children's learning and lives, the Internet has also brought about social problems [25]. Among such social problems, Internet addiction or screen time has received considerable attention [26, 27]. The use of Internet media can harm all aspects of children's body functions: It causes eye fatigue and reduces vision levels; Prolonged video screen time will put children in a "sedentary" state, squeezing out time for outdoor activities, quickly leading to obesity and a lack of motor skills; Using electronic products so that children are often crouching or lying on the bed or sofa affects children's cervical vertebrae and lumbar vertebrae, causing harm, which affects growth and development [28, 29].

Children's use of electronic devices is becoming increasingly popular, and children's poor self-control during this period may lead to electronic addiction if they are overexposed [30]. Related studies have also found that left-behind children spend more time on screens than the average child [31]. Although many people call for parents to return to their families to solve the problem of Internet addiction among left-behind youth, many parents are forced to work outside the home to make ends meet, and this situation is quite helpless but burdensome to change [32]. Cognitive characteristics and parenting patterns are more accessible to change than those of the left-behind situation [33]. Therefore, guardians play a vital role in preschool left-behind children's screen time. Guardians can also take measures to monitor, manage, and educate children's media use behavior in their daily lives to help children use media reasonably and correctly [34].

Studies have shown a correlation between visual screen time, physical activity, and motor skills in children [35]. Unreasonable screen time for children and adolescents will hurt their physical and mental health, such as poor vision, obesity, sleep disorders, anxiety, Internet addiction, decreased physical activity, and impaired language development [36, 37]. Physical activity levels are reported to be low in preschool children [38, 39]. Generally, preschool children engage in high amounts of sedentary behavior and low physical activity. The rise in preschoolers' screen time impacts their physical activity. Studies have shown that the level of gross motor movement in preschool children is directly related to physical activity [40]. However, the importance of gross motor movement is often overlooked by guardians who generally need more knowledge about parenting related to children's need for extensive physical activity. Based on the existing literature, the following hypotheses were proposed: There is a negative correlation between screen time and gross motor movement levels among preschool left-behind children.

Therefore, this study investigated the current screen time and gross motor movement of left-behind children aged 3–6 and identified the correlation between both. This study is intended to provide a theoretical basis for guardians to address the importance of gross motor movement in left-behind children.

## Methods

### Study design

Data were collected from rural Xiangcheng city of Henan province to left-behind families settings. These settings have several challenges inherent to rural areas, including many left-behind families and poor educational outcomes. Rural preschools had minimal education levels with limited outdoor play equipment.

We applied a cross-sectional methods approach consisting of assessing preschoolers' gross motor movement and the use of questionnaires to investigate preschoolers' screen time. This

study was checked concerning the STROBE (The Strengthening the Reporting of Observational Studies in Epidemiology) checklist [41]. From September to November 2021, one kindergarten was randomly selected from each of the 15 townships in Xiangcheng City, Henan province, using the whole-group random sampling method.

Inclusion criteria for the sample were: (1) being a student in preschool (in China, children aged 3–6 need to attend preschools and start elementary school at age 7); (2) absence of the disorders (signals of a medical condition, genetic, psychiatric or neurological disorder, or physical disability); (3) an adequate IQ for their chronological age (mentally handicapped). All data collectors participated in a two-day training workshop facilitated at the University of Zhengzhou before data collection. This training included measurement and practical sessions to ensure the correct administration of the data collection instruments.

The study was approved by the Research Ethics Committee of the first author's institution. Informed consent was obtained from the participants and school administrator before the data collection. A written informed consent was obtained from each participant before enrolling. This study was consent from the left-behind children's guardian, and participants were informed that their participation was completely voluntary and that they could terminate it at any time.

## Participants

To test-retest reliability 60 children were randomly chosen and retested again between 7 days from the first assessment by the same evaluators. After seven days, the questionnaire was administered to the subject children and retested to the population above. A two-way mixed intraclass correlation coefficient was used to understand the test-retest reliability. The coefficients of the retest reliability vary between 0 and 1 (below 0.50 = poor; between 0.50 and 0.75 = moderate; between 0.75 and 0.90 = good; above 0.90 = excellent) [42, 43]. The coefficients of the retest reliability of the gross motor development and screen time instruments reached 0.673 and 0.628, respectively, which proved that the measurement had moderate internal consistency and moderate structural reliability.

The G*Power 3.1 program was used to determine the sample size and power. A minimum of 352 samples were reached for models with an alpha level of 0.05 and a power level of 0.80. This many participants can be considered sufficient for the sample size in the current study [44]. Then, 60 preschool children aged 3–6 years were randomly selected from each of the 15 kindergartens as the study subjects. Consent forms were distributed to parents with kindergarten teachers' assistance after obtaining the kindergarten director's consent.

A total of 900 consent forms and questionnaires were distributed. To improve the data's authenticity, the questionnaires' quality was examined, and after the exclusion of invalid questionnaires (such as multiple answers for the same item and regular answers), there were 817 remaining valid questionnaires (validity rate = 90.7%). Of the respondents, 441 were boys (53.98%), and 376 were girls (46.02%); 107 were 3-year-olds (13.09%), 272 were 4-year-olds (33.29%), 242 were 5-year-olds (29.62%), and 196 were 6-year-olds (24%), the subjects' ages ranged from 3 to 6 years old (Mage = 4.37±0.87 years).

## Measures

**Test of Gross Motor Development (TGMD-3).**   This TGMD-3 was developed by Dale (2019) and is used to assess the gross motor development of children aged 3 to 10 [45]. The 13-item scale includes two dimensions: separate locomotor (running, galloping, hopping, skipping, horizontal jumping, and sliding) and ball skill subtests (two-handed and one-handed striking, dribbling, overhand throwing, underhand throwing, catching, and kicking). The

locomotor subtest has a total score of 46, and the ball skills subtest has a total score of 54. Each student performed the test items across two trials, each scoring based on 3 to 5 specific performance criteria depending on the testingitem (0 = did not perform correctly; 1 = performed correctly). The scores for each skill test item can range from 0 to 6, 0 to 8, or 0 to 10, depending on the number of performance criteria. Data used for analysis included the locomotor subtest score, the ball skills subtest score, a gross motor test score (i.e., locomotor subtest score + ball skill subtest score), and each two-trial total skill score.

Four trained graduate students in physical education scored each movement according to whether it met the criteria specified in the entry. The scores were summed to obtain a total gross score on the scale, with higher scores indicating higher levels of gross motor development. This test has been shown to have high reliability and validity in various cultural settings. It can be used to evaluate the development of gross motor movement in Chinese children [46]. The mean and standard deviation of the gross motor indexes were standardized, and the standardized Z-score of each index was calculated by the formula [Z-score = (test value—mean) / standard deviation].

**Screen time.**    The questionnaire was handed over to the guardian by the kindergarten teachers at the end of the school day and collected at the beginning of the next school day's class. There are five school days and two weekend days in total. The daily screen time information was collected by "the average number of minutes of screen time per day in the past 1d, except for class time", including the total time spent on TV, cell phone, and desktop computer. The WHO recommends that preschool children do not spend more than two hours of cumulative screen time per day, and more than two hours is considered excessive screen time [47–49]. Therefore, screen time was divided into two groups: more than 2h/d and less than or equal to 2h/d.

The screen time questionnaire was collected in two parts: during the kindergarten entry period, the classroom teacher was responsible for recording the children's screen time, and outside of kindergarten (outside of kindergarten on working days and days off), the parents recorded the children's screen time. Through the kindergarten parents' meeting, the purpose of the study and the process of completing the questionnaire were explained in detail to all parents. After obtaining parents' informed consent, an online parent chat group was set up to guide children's parents in completing the questionnaire. The researcher checked the questionnaires' completion daily, reminded parents of the importance and authenticity of the questionnaires, and answered their questions.

## Statistical analysis

The measured data were statistically analyzed using SPSS version 25.0. Firstly, an assumption test is performed to verify that the normality assumption is satisfied. One-way ANOVA was conducted to test for age and gender differences in the subjects' gross motor development scores and screen time. Correlations between displacement ability, object transfer ability, balance ability, gross motor development, and screen time were analyzed using Pearson's correlations for subjects of different ages. Finally, multilayer regression analysis was conducted by controlling for age and gender variables, with screen time as the independent variable and total gross motor development score as the dependent variable. The assumption test was verified before conducting the analysis. The test level P = 0.05.

## Results

### Assumption test

To test the assumption of normal distribution, Skewness should be within the range ±2; Kurtosis values should be within the range of ±7. The normality test results showed that the skewness

**Table 1. Average daily video screen time for 1 week for left-behind children aged 3 to 6 years.**

|  | Age | N | Screen Time | Number of detections | χ² value | P value |
|---|---|---|---|---|---|---|
| Overall | 3~ | 107 | 1.17±0.82 | 23 (21.49%) | 8.32 | 0.002** |
|  | 4~ | 272 | 1.39±1.16 | 85 (31.25%) |  |  |
|  | 5~ | 242 | 1.58±1.03 | 77 (31.82%) |  |  |
|  | 6~ | 196 | 1.66±1.26 | 61 (31.12%) |  |  |
| Boys | 3~ | 66 | 1.22±1.06 | 14 (21.21%) | 4.56 | 0.043* |
|  | 4~ | 148 | 1.46±1.13 | 52(35.14%) |  |  |
|  | 5~ | 123 | 1.61±1.25 | 42 (34.15%) |  |  |
|  | 6~ | 104 | 1.71±1.33 | 38 (36.54%) |  |  |
| Girls | 3~ | 41 | 1.13±0.84 | 9(21.95%) | 6.81 | 0.011** |
|  | 4~ | 124 | 1.32±1.07 | 33 (26.61%) |  |  |
|  | 5~ | 119 | 1.55±1.08 | 35 (29.41%) |  |  |
|  | 6~ | 92 | 1.62±1.24 | 23(25%) |  |  |

Note. figures in () are composition ratios or detection rates %; N: number of subjects

*p<0.05

**p<0.01

coefficients of the data ranged from -0.732 to -0.263, and the kurtosis coefficients ranged from -0.579 to 0.615, which were consistent with the premise of the normality assumption.

## Screen time (>2h/d) of left-behind children aged 3 to 6 years old by gender

As shown in Table 1, the average daily screen time of left-behind children aged 3–6 years increased gradually with age during one week, and the difference in the detection rate of >2h/d screen time was statistically significant ($p < 0.05$). The difference in the detection rate of >2h/d screen time of left-behind children aged 3–6 years was also statistically significant ($p < 0.05$) for different genders, respectively.

## Gross motor development characteristics of left-behind children aged 3 to 6 years

As shown in Table 2, the total gross motor movement score, separate locomotor score, and ball skill score of children aged 3 to 6 increased significantly with age, and the differences were statistically significant (F = 10.782, 5.293, and 7.688; P<0.01).

**Table 2. Gross motor movement characteristics of left-behind children aged 3 to 6 years.**

| Age | N | Locomotor |  | Ball skill |  | Gross motor movement |  |
|---|---|---|---|---|---|---|---|
| 3~ | 107 | 20.51±4.66 |  | 23.00±5.43 |  | 43.52±10.23 |  |
| 4~ | 272 | 25.97±5.40 |  | 29.17±3.76 |  | 55.14±7.95 |  |
| 5~ | 242 | 28.47±5.12 |  | 32.58±3.07 |  | 61.06±9.28 |  |
| 6~ | 196 | 32.59±4.78 |  | 30.18±3.64 |  | 62.77±8.18 |  |
|  |  | F-value | 5.293 | F-value | 7.688 | F-value | 10.783 |
|  |  | P- value | 0.000*** | P-value | 0.000*** | P-value | 0.000*** |

Note.

***p<0.001

### Analysis of the correlation between gross motor development and screen time in left-behind children aged 3 to 6 years

As shown in Table 3, there were significant negative correlations between total gross motor movement scores, separate locomotor scores, ball skill scores, and screen time for the left-behind children in each age group from 3 to 6 years old (P<0.05).

Given the significant correlation between gross motor movement and screen time in left-behind children aged 3–6 years, further investigate the effect of screen time on gross motor movement. This study used screen time as the dependent variable, age, and gender as independent variables in the first model, and gross movement total score in the second model for multiple linear regression analysis. As shown in Table 4 The total screen time score was predictive of the gross motor movement total score (adjusted $R^2$ = 0.184, p<0.001), with an explanatory rate of 21.4%. Specifically, for every 1-unit change in the total screen time score, the total gross motor movement score changes by -0.689 standard deviations.

## Discussion

With China's economic and social development, many rural parents go out to work. They leave their children behind in their hometowns and entrust their children to their elderly grandparents to raise. As a result, a large group of left-behind children has been formed in China. For a long time, the government and society have focused on caring for left-behind children to ensure they are well-fed and well-clothed. With smartphones and mobile Internet popularity in rural areas, increased screen time is becoming a new problem. It is urgent to implement "source management" by the supervisory department. At the same time, the cooperation of schools, families, and society is needed to create a safe and healthy space for left-behind children to grow up. The left-behind children were chosen for this study because the lack of parental roles significantly impacts the environment in which left-behind children grow up, making this population a particular risk group for early childhood development.

**Table 3. Analysis of the correlation between gross motor movement and screen time among left-behind children aged 3 to 6 years.**

|  | Age | Locomotor | Ball skill | Gross motor movement |
|---|---|---|---|---|
| Screen Time | 3~ | -0.354** | -0.162* | -0.164* |
|  | 4~ | -0.258* | -0.195* | -0.133* |
|  | 5~ | -0.318** | -0.182* | -0.212* |
|  | 6~ | -0.234* | -0.174* | -0.241* |

Note.

*p<0.05

**p<0.01; TGMD-3:Test of Gross Motor Development(3rd)

**Table 4. Multiple regression analysis of screen time and gross motor movement among left-behind children aged 3 to 6 years.**

|  | Predictors | B | SE | β | t | p | $R^2$ | AdjR$^2$ | $R^2$ changes |
|---|---|---|---|---|---|---|---|---|---|
| Gross motor movement | Age | 1.529 | 0.282 | 0.326 | 4.796 | 0.000*** | 0.138 | 0.184 | 0.214 |
|  | Gender | -1.364 | 0.475 | -0.051 | -1.391 | 0.173 |  |  |  |
|  | Screen Time | -0.758 | 0.041 | -0.689 | -10.783 | 0.000*** |  |  |  |

Note

***p < 0.001. B: unstandardized beta; SE: standard error for the unstandardized beta; β: standardized beta or standardized regression coefficient; t: t-statistic value; p: p-value of t-statistic; AdjR$^2$: adjusted R-squared

Studies have shown that the lack of family management is the main reason for increased screen time for left-behind children [50]. Guardians of left-behind children cannot restrain left-behind children from using cell phones, tablets, or watching TV. Guardians need to do housework or other tasks while looking after children, making it challenging to keep an eye on them throughout the day. Guardians use cell phones and tablets as "babysitters" so the children do not run around and disturb themselves. As a result, left-behind children have the power to dominate their phones or tablets too early and become attached to the online world from an early age.

On the other hand, some studies have confirmed that the decline in children's physical fitness is closely related to physical inactivity [51]. Physical inactivity among children has become a global universal problem, especially among children left behind in rural than urban areas [52]. Related research confirms that children with higher screen time spend less time engaged in TPA (Total of Physical Activity), MVPA (Moderate-to-Vigorous Physical Activity), and VPA, and more time in static behavior (chiefly screen-time behavior) [53]. Reducing static behavior and increasing physical activity has become a common demand for public health and physical education efforts. Regular physical activity can have several health-promoting effects on preschoolers, including promoting growth and enhancing gross motor movement levels [54, 55]. Therefore, this study was conducted to investigate whether there was a correlation between screen time and gross motor movement in left-behind children aged 3 to 6 years old and to analyze the differences in the characteristics of screen time and gross motor movement in left-behind children aged 3 to 6 years old.

This study showed that the development of locomotor ability, ball skill ability, and gross motor movement of left-behind children in each age group differed significantly with age and showed an overall increasing trend, which was consistent with the results of other prior studies [56]. It has been shown that children aged 3–6 are at a critical time in developing motor skills [57]. Suppose they are not given adequate stimulation during this period. In that case, it will affect the learning of motor skills and the physical application of complex movements later in life [58]. Therefore, in the motor learning of 3-6-year-olds, emphasis can be placed on learning through gross motor movement and exploring their surroundings to learn motor skills. Fundamental motor skills and various basic motor skills must be developed during preschool.

This study found that the average daily screen time increased with age among left-behind children aged 3–6 years, with significant differences between ages, and the reported rates of >2 h/d ranged from 22.43% to 33.73%. Cyberspace fits with curiosity and children's nature of liking games, being spontaneous, unrestrained, and chasing pleasure. Children need to judge the Internet correctly; they are only interested in sound and images, so games and animation become their only knowledge of cell phones, computers, and the Internet. The amount of screen time for children aged 3 to 6 years is related to the child's age; usually, the older the preschooler, the longer the screen time. Studies have shown that too few restrictions on children's video screen content can hurt children's play, hobbies, sleep hygiene, and eating habits [59].

Most guardians were unaware of unhealthy viewing and the associated deleterious effects. Past studies have shown that excessive screen time can lead to poor posture, scoliosis, impairment of motor coordination, visual perceptual development, cognitive brain development, poor concentration and creativity, behavioral deviations, and physical health, affecting many children with poor physical fitness [60]. Although excessive screen time has some negative impacts on children, as children grow, a one-size-fits-all approach is not advisable [61]. Guardians must decide how much media to let children use daily and what is appropriate. Establish clear rules and set reasonable limits for child's use of digital media [62]. For example, for children ages 3 to 6, limit screen time to one hour a day of high-quality programming (videos that are beneficial to children's development, such as educational videos, enlightenment

videos, children's dance videos, and children's song videos). Guardians will likely need to continue to guide, manage, and monitor a child's use of screens and media as he or she grows.

The present study showed that age was a significant predictor of gross motor development in children aged 3–6 under multivariate analysis, and screen time was a predictor of gross motor movement level in children aged 3–6. This is because the development of the movement in the preschool years is almost simultaneous with physical, emotional, and cognitive development. Significantly, the younger the age, the closer the relationship [63, 64]. Lack of experience with multiple basic motor skills in children 3–6 will negatively affect their later learning of fine or higher-level motor skills or cause difficulties in subject learning [65]. Therefore, motor development is not only related to the individual's physical and mental growth and development but also to the individual's future performance in learning and social interaction. In recent years, studies have found that children aged 3–6 lack time for physical activity, and 3C electronic devices such as TVs and cell phones limit children's opportunities for physical activity, leaving children at a critical stage of their development with a dilemma in learning and motor development [66, 67]. The increase in screen time has led to more static activities and a decrease in physical activity with age, which is a situation that needs to be observed and improved.

The 2018 Guidelines for Exercise in Preschool Children were released, which state that it is essential to actively foster a lifestyle where exercise is part of children's lives and stays with them throughout their lives between the ages of 3 and 6. The guidelines state that preschoolers should be active throughout the day to promote growth and development and that guardians should encourage children to play actively and should not exceed one hour of screen time. Therefore, it is recommended that guardians of left-behind children should limit children's video screen behavior promptly and should be able to promote the health of children and adolescents with at least 60 minutes of moderate physical activity daily to enhance the level of gross movement of left-behind children aged 3 to 6 years [68].

## Limitations

This study has several limitations that should be addressed. First, the data were cross-sectional, so no causality conclusions may be drawn. This is an initial investigation into the interrelationship of screen time and gross motor movement, but longitudinal work and experimental designs are essential for examining directionality and causality. Second, this study did not classify the left-behind children in detail, and future studies may consider selecting more targeted left-behind children for research or conducting stratified research, such as research on children left behind in different situations, such as single left-behind (One of the parents works outside the home all year round) and double left-behind (Parents work outside the home all year round). Third, some relevant variables are missing. For example: the content of what the children watch on the screens is not considered. Fourth, the retest reliability of the measurement is below 0.7, which is a shortcoming of the study, and subsequent studies will use measures higher than 0.7. Fifth, teachers teach according to the school's teaching plan, but due to the limited researchers, there are some difficulty in checking the daily screen usage with each teacher. Another strong limitation in the analysis was the lack of verification of screen time in the day care center by the evaluated children, and subsequent studies will improve this limit point.

## Implication

Investigating the relationship between the left-behind children's screen time and gross motor movements helps clarify the influence mechanism of the left-behind children's gross motor

movements. It provides a reference for the study of gross motor movements of left-behind children in this field. It is beneficial to propose targeted intervention methods of motor skill intervention for left-behind children from the perspectives of pedagogy and kinesiology. Further, it enlightens the research in early childhood physical education.

In-depth research in the area of motor development will bring benefits to the healthy growth of over 9 million left-behind children in China. The research on motor development of left-behind children aged 3 to 6 years old not only enriches the theory of motor development but also provides a good foundation for subsequent physical education. It provides good theoretical guidance for scientifically evaluating left-behind children's growth status and motor development.

Several practical implications have been derived from this study that warrant further investigation. First, age differences were present in screen time and gross motor movement in left-behind children aged 3 to 6 years old. This finding underlies a vital time during childhood to target interventions that may minimize these health-related gaps, specifically by decreasing screen time and focusing on fine gross movement. Second, screen time was well above the recommendations for this age group. Gross motor movement proficiency components were inversely related to more excellent screen time. For a long time, "parents going out to work" has become a potential threat to the education of "left-behind children," it is vital to help these children grow up healthily. With the development of information society and the popularization of Internet technology, the chance of left-behind children's access to electronic products such as TV and cell phones has increased, which makes left-behind children's screen time higher. Therefore, it is necessary to conduct regular training to deepen the guardians' understanding of the importance of video screen time and gross motor movements and to help them master physical education skills.

## Conclusion

There is a correlation between screen time and gross motor movement development in children aged 3–6 years old left behind, and the gross motor movement ability of children aged 3–6 years old can be developed by reducing screen time and increasing physical activity. This study can be used to inform teachers and families of the association between levels of screen time and the gross motor movement ability of children aged 3–6 years old. We suggest that guardian should guide and supervise children to reduce the screen time and encourage more verbal contact with the environment, to reduce damage to gross motor coordination.

## Author Contributions

**Methodology:** Pengwei Song, Long Qin.

**Writing – original draft:** Rui Yuan.

**Writing – review & editing:** Jia Zhang.

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
