## [Decision Letter · Decision Letter 0]

4 Oct 2023

PONE-D-23-18301The relationship between screen time and gross motor movementPLOS ONE

Dear Dr. Zhang

Thank you for submitting your manuscript to PLOS ONE. After careful consideration, we feel that it has merit but does not fully meet PLOS ONE’s publication criteria as it currently stands. Therefore, we invite you to submit a revised version of the manuscript that addresses the points raised during the review process.

We look forward to receiving your revised manuscript.

Kind regards,

Tadashi Ito

Academic Editor

PLOS ONE

Journal Requirements:

3. Please amend either the title on the online submission form (via Edit Submission) or the title in the manuscript so that they are identical

Reviewers' comments:

Reviewer's Responses to Questions

**Comments to the Author**

1. Is the manuscript technically sound, and do the data support the conclusions?

Reviewer #1: Partly

Reviewer #2: Yes

2. Has the statistical analysis been performed appropriately and rigorously? 

Reviewer #1: No

Reviewer #2: Yes

3. Have the authors made all data underlying the findings in their manuscript fully available?

Reviewer #1: No

Reviewer #2: Yes

4. Is the manuscript presented in an intelligible fashion and written in standard English?

Reviewer #1: No

Reviewer #2: Yes

5. Review Comments to the Author

Reviewer #1: 1. The text needs to have an English language and textual review. In the introduction there is a lot of repetition of phrases with the same meaning. It can be more summarized.

2. The theme is very interesting, bringing the profile of an unusual and "official" population of child care, the so-called "left-behind children".

3. It was not clear in the consent form, how the parents were contacted to sign the participation of their children, under 36 years old, to participate in the application of the instrument. The amount is very high and this form of contact needs to be made clear. 4. I was surprised that a study that involved so many families and children did not have financial support for its realization.

4. The study is cross-sectional and, as mentioned, it cannot analyze the causality between whether the use of the screen is what causes the consequence in the gross motor pattern, in the child's development. However, I noticed that there was no separation by age, having grouped all the children in the analysis.

5. Was a separate analysis made between the different ages, the time in which parents are away from their children?

6. How did the authors assess the quality of the information recorded by parents?

7. Did the authors ask the question about the type of use of the cell phone (who owns it and if the parents control the time of use and observe the type of program watched)? In this case, as they are very young children, it is the parents who give them access to the screen.

8. As it is a lower-income population, how many types of screens do children have access to at home? Was there variation between the time spent using the cell phone/tablet or TV? It is not clear from the text.

9. In the discussion item, I considered that there was a limited search for updated references on the subject, which have been published in excellent journals, including pointing out differences between types of exhibitions

10. Another strong limitation in the analysis was the lack of verification of screen time in the day care center by the evaluated children.

11. It was not clear in the text the details of how the instrument was applied and how the parents were interviewed by the four field researchers. Detailing this whole process is important

12. In the conclusions, the authors should mention what kind of proposal they indicate for the participation of parents, in an intervention that seeks to guide them to reduce screen time and encourages more verbal contact with the environment, to reduce damage to gross motor coordination.

13. The text could be more concise, but clearly showing the steps of the research, which can be reproduced by other researchers in the countries mentioned that it contains many "children left behind".

14. It was not clear in the study design how the sample that would be representative of this group of children was selected

Reviewer #2: The authors performed a study addressing “The relationship between screen time and gross motor movement”. However, several concerns to be clarified, some of which are critical.

1. The Introduction seems a bit long. Can you summarize the main points and provide more Introduction?

2. The background factors of the subject are very important. It changes the results. However, I think there is not enough description of background factors. Please describe them.

3. On p. 18, lines 338-339, "The results are consistent with previous studies (Schmidt et al., 2020)." does this mean that the results were the same for left-behind children? Please correct this as it is unclear.

6. PLOS authors have the option to publish the peer review history of their article (what does this mean?). If published, this will include your full peer review and any attached files.

Reviewer #1: No

Reviewer #2: No

---

## [Author Response · Author response to Decision Letter 0]

18 Nov 2023

Dear reviewer: 

Thank you very much for your meticulous review. Here are my answers to your questions and the relevant revisions to the article. The modifications have been highlighted in blue.

(i)The text needs to have an English language and textual review. In the introduction there is a lot of repetition of phrases with the same meaning. It can be more summarized.

The full text has been reviewed and revised.

(ii)The theme is very interesting, bringing the profile of an unusual and "official" population of child care, the so-called "left-behind children".

Left-behind children are commonly found in China or Southeast Asian countries and are a special group of people worthy of attention, so we have chosen them as research objects.

(iii)It was not clear in the consent form, how the parents were contacted to sign the participation of their children, under 36 years old, to participate in the application of the instrument. The amount is very high and this form of contact needs to be made clear. 

We contact their guardians through the kindergarten. 

(iv)I was surprised that a study that involved so many families and children did not have financial support for its realization.

It has now been added.

(v)The study is cross-sectional and, as mentioned, it cannot analyze the causality between whether the use of the screen is what causes the consequence in the gross motor pattern, in the child's development. However, I noticed that there was no separation by age, having grouped all the children in the analysis.

Age distribution has been added in Participants section.

(vi)Was a separate analysis made between the different ages, the time in which parents are away from their children?

This study did not analyze the departure time of parents of left-behind children. In follow-up research, we will further analyze variables such as the time when parents leave for children of different ages, whether both parents leave or one parent leaves.

(vii)How did the authors assess the quality of the information recorded by parents?

Through the kindergarten parents' meeting, the purpose of the study and the process of completing the questionnaire were explained in detail to all parents. After obtaining parents' informed consent, an online parent chat group was set up to guide children's parents in completing the questionnaire. The researcher checked the questionnaires' completion daily, reminded parents of the importance and authenticity of the questionnaires, and answered their questions.

(viii)Did the authors ask the question about the type of use of the cell phone (who owns it and if the parents control the time of use and observe the type of program watched)? In this case, as they are very young children, it is the parents who give them access to the screen.

The daily screen time information was collected by "the average number of minutes of screen time per day in the past 1d, except for class time", including the total time spent on TV, cell phone, and desktop computer. It is not just the amount of time children spend looking at cell phones but also the amount of time they spend watching TV.

Because smartphones have become popular in China, we did not investigate the type of phone. Guardians control the amount of time they spend on their cell phones to varying degrees (watching cartoons, mini-games, anime, etc.). But it's up to the guardians themselves. Many guardians use electronic products to keep their restless children quiet and avoid disturbing their time. Some parents even think that video games and videos can also promote the development of their children's intelligence and that playing them has no harm. However, they are oblivious to the dangers of their children's indulgence in electronic screens.

(ix)As it is a lower-income population, how many types of screens do children have access to at home? Was there variation between the time spent using the cell phone/tablet or TV? It is not clear from the text.

Smartphones and TVs are so cheap in China that they have long been popularized, and almost every family has TV and smartphone. Left-behind children are not all from low-income families. The phenomenon of parents leaving their children to go to work is common in China's towns and villages.

There is no difference between time spent using a cell phone/tablet or TV. We define screen time as including the total time spent on TV, cell phone, and desktop computer.

(x)In the discussion item, I considered that there was a limited search for updated references on the subject, which have been published in excellent journals, including pointing out differences between types of exhibitions

The references in the discussion section have been checked and updated.

(xi)Another strong limitation in the analysis was the lack of verification of screen time in the day care center by the evaluated children.

It has been added to the text. “Fifth, teachers teach according to the school's teaching plan, but due to the limited researchers, there are some difficulty in checking the daily screen usage with each teacher. Another strong limitation in the analysis was the lack of verification of screen time in the day care center by the evaluated children, and subsequent studies will improve this limit point.”

(xii)It was not clear in the text the details of how the instrument was applied and how the parents were interviewed by the four field researchers. Detailing this whole process is important.

It has been added to the text. Through the kindergarten parents' meeting, the purpose of the study and the process of completing the questionnaire were explained in detail to all parents. After obtaining parents' informed consent, an online parent chat group was set up to guide children's parents in completing the questionnaire. The researcher checked the questionnaires' completion daily, reminded parents of the importance and authenticity of the questionnaires, and answered their questions.

(xiii)In the conclusions, the authors should mention what kind of proposal they indicate for the participation of parents, in an intervention that seeks to guide them to reduce screen time and encourages more verbal contact with the environment, to reduce damage to gross motor coordination.

It has been added to the text. “This study can be used to inform teachers and families of the association between levels of screen time and the gross motor movement ability of children aged 3-6 years old. We suggest that guardian should guide and supervise children to reduce the screen time and encourage more verbal contact with the environment, to reduce damage to gross motor coordination.”

(xiv)The text could be more concise, but clearly showing the steps of the research, which can be reproduced by other researchers in the countries mentioned that it contains many "children left behind".

The investigation steps have been supplemented.

(xv)It was not clear in the study design how the sample that would be representative of this group of children was selected.

Henan is a large province of left-behind children; as of 2021, 1,595,600 rural left-behind children in compulsory education are in school, accounting for 10.70% of the total number of students in compulsory education, excluding preschool left-behind children (More than 120,000) . Henan Province has large -scale left -behind children, so we choose a city in Henan Province for investigation and research to ensure the representative of the sample.

Dear reviewer: 

Thank you very much for your meticulous review. Here are my answers to your questions and the relevant revisions to the article. The modifications have been highlighted in blue.

(i)The Introduction seems a bit long. Can you summarize the main points and provide more Introduction?

We modified the introduction part.

(ii)The background factors of the subject are very important. It changes the results. However, I think there is not enough description of background factors. Please describe them.

Added related content to the text. “Studies have shown a correlation between visual screen time, physical activity, and motor skills in children (André & Cochetel, 2022; Chen et al., 2022b). Unreasonable screen time for children and adolescents will hurt their physical and mental health, such as poor vision, obesity, sleep disorders, anxiety, Internet addiction, decreased physical activity, and impaired language development (Thomas et al., 2020; Luiza et al., 2023). Physical activity levels are reported to be low in preschool children (Neville et al., 2021; Gao et al., 2022; Wang et al., 2022). Generally, preschool children engage in high amounts of sedentary behavior and low physical activity. The rise in preschoolers' screen time impacts their physical activity. Studies have shown that the level of gross motor movement in preschool children is directly related to physical activity (Yang et al., 2022a). However, the importance of gross motor movement is often overlooked by guardians who generally need more knowledge about parenting related to children's need for extensive physical activity.”

(iii)On p. 18, lines 338-339, "The results are consistent with previous studies (Schmidt et al., 2020)." does this mean that the results were the same for left-behind children? Please correct this as it is unclear.

Related error sentences have been deleted.

---

## [Decision Letter · Decision Letter 1]

20 Dec 2023

The relationship between screen time and gross motor movement:A cross-sectional study of pre-school aged left-behind children in China

PONE-D-23-18301R1

Dear Dr. Jia Zhang,

We’re pleased to inform you that your manuscript has been judged scientifically suitable for publication and will be formally accepted for publication once it meets all outstanding technical requirements.

Kind regards,

Tadashi Ito

Academic Editor

PLOS ONE

Additional Editor Comments (optional):

Reviewers' comments:

Reviewer's Responses to Questions

**Comments to the Author**

1. If the authors have adequately addressed your comments raised in a previous round of review and you feel that this manuscript is now acceptable for publication, you may indicate that here to bypass the “Comments to the Author” section, enter your conflict of interest statement in the “Confidential to Editor” section, and submit your "Accept" recommendation.

Reviewer #1: All comments have been addressed

Reviewer #2: (No Response)

2. Is the manuscript technically sound, and do the data support the conclusions?

Reviewer #1: Yes

Reviewer #2: Partly

3. Has the statistical analysis been performed appropriately and rigorously? 

Reviewer #1: Yes

Reviewer #2: Yes

4. Have the authors made all data underlying the findings in their manuscript fully available?

Reviewer #1: Yes

Reviewer #2: Yes

5. Is the manuscript presented in an intelligible fashion and written in standard English?

Reviewer #1: Yes

Reviewer #2: Yes

6. Review Comments to the Author

Reviewer #1: The authors reviewed the observations highlighted in the opinion and reformulated some paragraphs, making them more direct and clear. The topic is of great relevance, as it is one of the biggest public health problems worldwide. I believe that this publication can encourage parents and public managers to adopt more targeted interventions to reduce screen time. I suggest that authors seek to carry out follow-up studies in this population or use randomized models to develop long-term cohort studies.

Reviewer #2: (No Response)

7. PLOS authors have the option to publish the peer review history of their article (what does this mean?). If published, this will include your full peer review and any attached files.

Reviewer #1: No

Reviewer #2: No

---

## [Editor Report · Acceptance letter]

31 Jan 2024

PONE-D-23-18301R1 

PLOS ONE

Dear Dr. Zhang, 

I'm pleased to inform you that your manuscript has been deemed suitable for publication in PLOS ONE. Congratulations! Your manuscript is now being handed over to our production team.

Kind regards, 

on behalf of

Dr. Tadashi Ito 

Academic Editor

PLOS ONE